molecular biology/biotechnology

Chinese kale, phytoene desaturase, light, phytohormone, subcellular localization, CRISPR/Cas9

**Authors for correspondence:**
Fen Zhang
e-mail: zhangf_12@163.com
Hao-Ru Tang
e-mail: htang@sicau.edu.cn

# Functional differences of *BaPDS1* and *BaPDS2* genes in Chinese kale

Bo Sun[1,†], Min Jiang[1,†], Sha Liang[1,†], Hao Zheng[1], Qing Chen[1], Yan Wang[2], Yuan-xiu Lin[2], Ze-Jing Liu[1], Xiao-Rong Wang[2], Fen Zhang[1] and Hao-Ru Tang[1,2]

[1]College of Horticulture, and [2]Institute of Pomology and Olericulture, Sichuan Agricultural University, Chengdu 611130, People's Republic of China

BS, 0000-0003-3306-656X; H-RT, 0000-0001-6008-2747

This study presents a systematic analysis of the functional differences between two genes that encode phytoene desaturase (PDS) in Chinese kale. The promoter sequences of both *BaPDS1* and *BaPDS2* were amplified and cloned, and their lengths were 2005 bp and 2000 bp, respectively. The mining of *cis*-acting elements in the promoters showed that the two *BaPDS* genes are mainly associated with light and phytohormone responsiveness. Light quality, light intensity and plant hormone treatments were conducted in seedlings of Chinese kale, and the results indicated that the response of the two genes to different factors differed. Among them, BaPDSs collectively respond to the treatment with salicylic acid and abscisic acid. With regard to response differences, *BaPDS1* is sensitive to red and blue light, blue light, and strong light, while *BaPDS2* responds to blue light, weak light, darkness, gibberellin and methyl jasmonate. In addition, both BaPDS1 and BaPDS2 are likely targeted to the chloroplast. Furthermore, single and double mutants of *BaPDSs* were generated via CRISPR/Cas9 technology. Phenotypic analysis showed that the double mutant with edited *PDS1* and *PDS2* was a pure albino, while the single mutants with edited *PDS1* or *PDS2* were partly whitened. In summary, *BaPDS1* and *BaPDS2* genes played different and indispensable roles in Chinese kale, and their functions were partially complementary.

## 1. Introduction

Carotenoids are isoprenoids that are widely found in both plants and microorganisms [1]. In plant photosynthesis, they can also be used as accessory pigments of the photosynthetic antenna, which can transmit visible energy to chlorophyll by capturing photons

†These authors contributed equally to this work.

and absorbing visible light [2]. In addition, carotenoids serve as precursors for the synthesis of abscisic acid (ABA) in higher plants [3]. The carotenoid biosynthesis pathway in higher plants consists of condensation of small molecular substances into C40 compounds, and a series of reactions such as dehydrogenation, isomerization, cyclization, hydroxylation and epoxidation to produce different carotenoids [1]. Phytoene desaturase (PDS) is one of the key rate-limiting enzymes in the carotenoid biosynthetic pathway [4], which catalyses the dehydrogenation of colourless phytoene to ζ-carotene [5].

The *PDS* gene in plants was first isolated from the mutants of *Phytophthora* in 1992 by Perker *et al.* [6]. In 1995, the antisense expression of the *PDS* gene was performed in *Nicotiana benthamiana* and a large amount of phytoene was found to accumulate in the leaves of this transgenic tobacco [7]. In 2000, the *PDS* gene was transferred into tomato, and the content of β-carotene increased about threefold in the fruit of transgenic tomato compared to the wild-type control [8]. At present, the *PDS* gene has been silenced in various plants, such as petunia, tobacco, *Arabidopsis*, cassava, cantaloupe, melon and cucumber by using the clustered regularly interspaced short palindromic repeats/CRISPR-associated proteins (CRISPR/Cas9) system or the virus-induced gene silencing (VIGS) system, which causes chlorophyll degradation, and leads to mottled, fading, yellowing and even whitening of chlorophyll-rich leaves and other tissues, as a result of photobleaching [9–11].

A gene family is a group of homologous genes present in two or more copies arising from gene duplication. The members of a gene family have significant similarities in both structure and function, encode similar protein products and can be closely arranged to form a cluster of genes. However, most of the time, they are scattered at different locations on the same chromosome, or exist on different chromosomes, each with different expression regulation patterns [12].

Chinese kale (*Brassica oleracea* var. *alboglabra*) is a vegetable of the *Brassicaceae*, which is cultivated all over China and is rich in nutrients such as glucosinolates, vitamin C and carotenoids [13]. The function of the *PDS* gene is well known, but studies on functional differences between different members of the *PDS* gene family have not been reported to date. In a previous study by our group, white-flower Chinese kale was used as plant material, and two members of the *BaPDS* gene family were isolated [14]. In the current study, based on the results of the *BaPDS* promoter *cis*-element analysis, light and phytohormones were activated in seedlings of Chinese kale to study the differences in expression patterns between both genes. Subsequently, subcellular localization of BaPDS proteins was performed, and both genes were edited using CRISPR/Cas9 technology to investigate functional differences in *BaPDS* gene family members.

# 2. Experimental method

## 2.1. Plant materials

The cultivar 'Sijicutiao' of white-flower Chinese kale was used in this study. The plants were grown in trays containing a mixture of peat and vermiculite (3 : 1) in an artificial climate chamber with a light intensity of $160 \, \mu\text{mol} \, \text{m}^{-2} \, \text{s}^{-1}$, a temperature of 25/20°C (day/night), a 12/12 h (day/night) light cycle, with humidity maintained between 70 and 80%. Fertilizer and water were applied as needed.

## 2.2. Light and phytohormone treatments

At an age of 30 days, Chinese kale seedlings that were growing under the conditions described in 2.1 were selected for light and phytohormone treatments. The used light qualities were red (R : B = 10 : 0), blue (R : B = 0 : 10), and red and blue (R : B = 5 : 5), and white light treatment was used as the control. Light intensities were dark ($0 \, \mu\text{mol} \, \text{m}^{-2} \, \text{s}^{-1}$), weak light ($80 \, \mu\text{mol} \, \text{m}^{-2} \, \text{s}^{-1}$), control ($160 \, \mu\text{mol} \, \text{m}^{-2} \, \text{s}^{-1}$) and strong light ($240 \, \mu\text{mol} \, \text{m}^{-2} \, \text{s}^{-1}$). The light source is directly above the plant, and the linear distance from light to the plants is about 15 cm. Phytohormones were sprayed on the leaf surface with 1 mM SA, 5 μM $GA_3$ and 0.5 μM ABA, respectively, and distilled water was used as the control. We stopped spraying until the leaf surface was dripping, and the amount was about 2 ml of each phytohormone per plant. In addition, 100 μM MeJA was used to fumigate the Chinese kale seedlings in a transparent and closed container [13]. When the leaves had stopped dripping, the plants were moved into an artificial climate chamber. The cultural conditions of the control and treated Chinese kale seedlings were as described in §2.1. The fifth to sixth true leaves were sampled 0, 1, 3, 6, 12, 24, 48 and 72 h after each treatment, respectively, and the samples were immediately frozen in liquid nitrogen and stored in a freezer at −80°C prior to RNA extraction.

**Table 1.** Primers used in this study.

| primer name | primer sequence (5′−3′) | annealing (°C) | aims |
|---|---|---|---|
| *PDS1*promoter-F | TCCAAGCTTTTGATCTGCTGCTTTTAAC | 56 | molecular cloning of |
| *PDS1*promoter-R | TCCTTCGCCTTGTTCTTGTCTTAAGC | | *BaPDSs* promoters |
| *PDS2*promoter-F | ATGCACTGAAGGTGATATTCATGCTTGC | 54 | |
| *PDS2*promoter-R | GCGGTGTTCCATCAAATCACAACC | | |
| *PDS1* qRT-PCR F | ATGGTTGTGTTTGGGAATGTTTCTGCA | 63 | detection of gene |
| *PDS1* qRT-PCR R | CCTGCAAAGGACAAGAAGTCCTTCG | | expression |
| *PDS2* qRT-PCR F | ATGGTTGTGTTTGGGAATGTTTCCGCG | 63 | |
| *PDS2* qRT-PCR R | TGCAAAGGACCAGCACTCCTCCT | | |
| *β-actin* qRT-PCR F | CCAGAGGTCTTGTTCCAGCCATC | 63 | |
| *β-actin* qRT-PCR R | GTTCCACCACTGAGCACAATGTTAC | | |
| *PDS1*-GFP-F | cgGGATCCATGGTTGTGTTTGGGAATGTTTCTGCA | 61 | detection of subcellular |
| *PDS1*-GFP-R | acgcGTCGACTGTTGATACAGTTGTCTCCGACAAG | | localization (vector) |
| *PDS2*-GFP-F | cgGGATCCATGGTTGTGTTTGGGAATGTTTCCGCG | 61 | |
| *PDS2*-GFP-R | acgcGTCGACTGTTGATAGAGTCGCCTCCGACAAC | | |
| *PDS*-CRISPR-F | CACCGATGGAGATTGGTATGAAAC | 55 | the synthesis of target site |
| *PDS*-CRISPR-R | AAACGTTTCATACCAATCTCCATC | | |
| *PDS1*-M-F | CCTGCAAAGCCTTTAAAAGTTGTCATT | 55 | detection of the mutation |
| *PDS1*-M-R | CCAAGTTCTCCAAATAAGTTCTGCACG | | in transgenic plants |
| *PDS2*-M-F | CCTGCAAAGCCTTTAAAAGTTGTGATC | 55 | |
| *PDS2*-M-R | GCTATAGAAGATAAGAGCCGAGCCT | | |

## 2.3. Genomic DNA and RNA extraction

Only healthy and disease-free true leaves of Chinese kale were sampled, and their genomic DNA was isolated using a rapid plant genomic DNA isolation kit (Sangon, Shanghai, China) according to the manufacturer's instructions. The obtained DNA was then used to amplify the promoter. Total RNA was extracted from true leaves of Chinese kale using an alternative CTAB method [15]. Intact total RNA was used for cDNA synthesis using the PrimeScript™ 1st Strand cDNA Synthesis Kit (TaKaRa, Japan). The cDNA was used to evaluate the expression levels of *BaPDSs* under diverse treatments via quantitative real-time PCR (qPCR).

## 2.4. Molecular cloning and *cis*-element analysis of *BaPDS* promoters

Using promoter sequences for PDS genes of homologous species such as cabbage and Chinese cabbage published by the NCBI, specific primers for promoters of the *BaPDS1* and *BaPDS2* were designed (table 1). The cDNA from true leaves was used as the template, and the PCR products were visualized on a 1% ethidium bromide-stained agarose gel. Prominent fragments were excised from the gel, purified and cloned into a pEASY-Blunt vector (TransGene, Beijing), and then sequenced by Sangon Biotech Co. Ltd (Shanghai, China). Subsequently, the *cis*-acting elements within the promoter sequences of *BaPDSs* were predicted using the PlantCARE online software (http://bioinformatics.psb.ugent.be/webtools/plantcare/html/).

## 2.5. Quantitative real-time PCR analysis

The gene expression of *BaPDSs* in Chinese kale leaves under different exogenous treatments was analysed by qPCR. According to the gene sequences of *BaPDS1* (GenBank Accession no.: KX426039) and *BaPDS2* (GenBank Accession no.: KX426040) previously isolated in our laboratory [14], fluorescence-specific primers for both genes were designed, and β-actin was used as the reference gene (table 1) [16]. The target

gene *BaPDSs* were amplified together with the reference gene β-actin using a Bio-Rad iCycler thermocycler (Bio-Rad, USA) in a total reaction volume of 20 μl. This volume contained 10 μl 2 × SYBR Premix EX Taq (Takara, Japan), 2 μl of each cDNA template and 1 μl of each gene-specific primer (10 μM). Each treatment used three biological replicates. Gene expression analysis was performed using the $2^{-\Delta\Delta CT}$ method [17].

## 2.6. Subcellular localization

Subcellular localization of *BaPDSs* was performed using the methods described by Sun *et al.* [18]. The complete CDS sequences of *BaPDS1 and BaPDS2* were amplified using the primers PDS1-GFP and PDS2-GFP (table 1), in which a *Bam*H I site at the 5′-end and a *Sal* I site at the 3′-end of the gene were incorporated, respectively. The expression vector pC2300-35S-eGFP and the amplification product were double-digested, and the digestion product was purified by a gel recovery kit. After this double digestion, the target genes were inserted into the empty vector, and the GFPs were in-frame fused at the C-terminal of *BaPDSs*. The high concentration recombinant plasmids pC2300-35S-*BaPDS1*-eGFP, pC2300-35S-*BaPDS2*-eGFP and empty pC2300-35S-eGFP were extracted using the SDS alkaline lysis method. Then, Chinese kale mesophyll protoplasts were isolated and purified, and the three above-mentioned recombinant plasmids were transformed into Chinese kale protoplasts according to the optimization protocols by Sun *et al.* [18]. The protoplasts expressing GFP fusion protein were observed and images were captured using a Zeiss AxioImager A2 fluorescence microscope (Carl Zeiss, Germany).

## 2.7. CRISPR/Cas9 editing of *BaPDSs*

A conserved target sequence of the *BaPDSs* was selected as the target site using CRISPR DESIGN (http://crispr.mit.edu/) and ZiFiT Targeter v. 4.2 (http://zifit.partners.org/ZiFiT/CSquare9 Nuclease.aspx) online software packages. The primers PDS-CRISPR (table 1) that were designed according to the sequence of the target site were annealed, renatured and inserted into the *Bbs*I cleavage site of the pSG vector. Then, the recombinant plasmid pSG-*BaPDSs* and pCC vector were double-digested using *Eco*RI-HF and *Xba*I, and the digested products were separately recovered and purified. Both recovered products were ligated to obtain the recombinant plasmid pCC-Target-sgRNA, which was confirmed by double digestion and sequencing. Then, the extracted plasmid was transformed into *Agrobacterium* by the freeze–thaw method.

The transformation of Chinese kale was performed using the methods of Qian *et al.* [19]. After the cotyledons were pre-incubated for 3 days, they were immersed in *Agrobacterium* liquid containing the recombinant plasmid pCC-Target-sgRNA for 1–2 min. The impregnated cotyledons were transferred into a co-culture medium and cultured for 5 days at 25°C in the dark, and then transferred to a selection medium containing 200 mg l$^{-1}$ timentin, 325 mg l$^{-1}$ carbenicillin and 12 mg l$^{-1}$ hygromycin. Subculture was performed every 20 days, and molecular identification and phenotypic analysis were performed after the adventitious buds reached a specific length.

The genomic DNA of the transgenic Chinese kale plants was extracted, and primers PDS1-M and PDS2-M were designed according to the target sequence. The genomic DNA was extracted from each transgenic Chinese kale plant and used as template. PCR amplification was conducted with primers PDS1-M and PDS2-M (table 1), and the amplified product of each transgenic plant was gel extracted, ligated, transformed and sequenced. Ten positive clones for each transgenic line were sent to analysis.

## 2.8. Colour

Colour analysis of the transgenic plants was conducted using an NR110 chromameter (3nh, Shenzhen). Three positions of each mutant plant were randomly selected, and the colour values were obtained as $L^*$, $a^*$ and $b^*$. The chromameter was calibrated via the standard white plate according to the manufacturer's instructions: $-L^*$ direction was black, $+L^*$ direction was white, $-a^*$ direction was green, $+a^*$ direction was red, $-b^*$ direction was blue and $+b^*$ direction was yellow. The $a^*$ or $b^*$ value was biased towards 0, which stands for white.

# 3. Results

## 3.1. Cloning the promoters of *BaPDSs*

PCR amplification was conducted using the extracted genomic DNA from Chinese kale 'Sijicutiao' as the template. After the products were subjected to agarose gel electrophoresis, two bands of about 2000 bp

were obtained, which was consistent with the predicted fragment size. Sequencing confirmed that both fragments were located upstream of the start codon (ATG) of *BaPDS1* and *BaPDS2*, and the sequence sizes were 2005 bp and 2000 bp, respectively, which were submitted to GenBank under accession numbers MK334183 and MK334184, respectively.

## 3.2. Analysis of *cis*-acting elements of *BaPDS* promoter sequences

The *cis*-acting elemental analysis of the *BaPDS* promoter sequences was performed with PlantCARE software. The results showed many light- and phytohormone-related elements within the *BaPDS* promoters, indicating the two genes may be regulated by light and hormones. Both promoters have multiple identical *cis*-acting elements, such as light-responsive elements (3-AF1 binding site, Box I, TCT-motif, etc.), and the salicylic acid (SA)-responsive element TCA-element (table 2). The promoters of *BaPDS1* and *BaPDS2* also contain many distinct *cis*-acting elements, including different types of light-responsive elements, in which the *BaPDS1* promoter has an AE-box, a GATA-motif and TCCC-motif elements, while the *BaPDS2* promoter has a GT1-motif, an AT1-motif, a GA-motif, a GAG-motif and ATCT-motif components. These also contain different phytohormone responsive elements: the *BaPDS1* promoter contains methyl jasmonate (MeJA) and ABA-responsive elements, while the *BaPDS2* promoter contains GA-responsive elements. In addition, the *BaPDS1* promoter also contains an Myb binding element that is involved in drought response (MBS) and a wound-responsive element, while the *BaPDS2* promoter also contains an Myb binding element that is involved in light response (MRE) (table 3).

## 3.3. Gene expressions of *BaPDSs* under light and phytohormone treatments

The results of the light quality treatment (figure 1) showed that the gene expression of *BaPDS1* in Chinese kale seedlings peaked at 3 h after blue and red and blue light treatments, and the expression decreased subsequently. However, the expression level of *BaPDS1* under red and blue light treatment was remarkably higher than that of the control, while that under blue light treatment showed no significant difference compared to the control after 3 h. The expression level of *BaPDS1* after red light treatment was not significantly different from that of the control. However, the gene expression of *BaPDS2* under blue light treatment was significantly lower than that of the control except at 1, 24 and 48 h, while the expression of *BaPDS2* under red and red and blue light treatment showed no obvious change compared to the control. These findings indicate that *BaPDS1* could be induced by red and blue and blue light, while *BaPDS2* could be suppressed by blue light.

The results of light intensity treatment showed that *BaPDS1* only responded to strong light treatment and its expression level peaked at 24 h, reaching 4.4-fold of that of the control, then declined rapidly. However, the expression level of *BaPDS2* reached its peak after 12 h under irradiation with weak light, then decreased subsequently, and matched the expression level of the control at 72 h. In addition, the expression level of *BaPDS2* under dark treatment was significantly lower than that of the control, and followed a gradually decreasing trend, showing no expression at 72 h. The expression of *BaPDS2* under strong light treatment was not significantly different than that of the control (figure 1). These findings indicate that *BaPDS1* could be induced by strong light, while *BaPDS2* could be induced by weak light and suppressed in the dark.

The expression levels of *BaPDS1* and *BaPDS2* were both suppressed after SA treatment. The expression level of *BaPDS2* was quickly induced and peaked at 1 h after gibberellin (GA$_3$) treatment, and was obviously higher under MeJA treatment than in the control treatment except at 12 h. However, the expressions of *BaPDS1* after GA$_3$ or MeJA treatments were similar to that of the control (figure 1). These findings indicate that *BaPDS1* and *BaPDS2* could both respond to ABA and SA, and *BaPDS2* also could respond to GA$_3$ and MeJA.

## 3.4. Subcellular localization of BaPDS proteins

Both BaPDS1 and BaPDS2 were predicted to be targeted into the chloroplast by the subcellular localization software WoLF PSORT. Localization to the chloroplast was confirmed by visualizing transient expression of BaPDS1 and BaPDS2 in Chinese kale mesophyll protoplasts using fluorescence microscopy. As shown in figure 2, clear GFP fluorescence signals of BaPDS1 and BaPDS2 were detected only in the chloroplast, and no GFP signal was found in other parts of the protoplast. These

**Table 2.** Common *cis*-acting elements in the promoter regions of *BaPDS1* and *BaPDS2*. + and − in parentheses represent sense strand and antisense strand, respectively. The same is as below.

| function | *cis*-element | Sequence (5'–3') | location of *BaPDS1* | location of *BaPDS2* |
| --- | --- | --- | --- | --- |
| light responsiveness | 3-AF1 binding site | AAGAGATATTT | −947(−) | −1268(−) |
| | Box I | TTTCAAA | −586(−) | −1951(+) |
| | TCT-motif | TCTTAC | −1938(+) | −1060(+) −1741(+) |
| | I-box | TATTATCTAGA | −475(−) | −1741(+) |
| | Box 4 | ATTAAT | −1932(+) −1200(−) −473(−) | −1080(−) −1142(−) |
| | G-box | CACGTT | −1536(+) −1605(+) −387(+) −1615(+) −390(−) | −1193(−) |
| salicylic acid responsiveness | TCA-element | CCATCTTTTT | −1788(+) −975(−) | −333(−) |
| anaerobic responsiveness | ARE | TGGTTT | −1287(−) | −193(−) −283(−) −1184(−) −554(+) |
| heat stress responsiveness | HSE | AAAAAATTTC | −1056(+) −1040(+) | −1254(+) |
| endosperm expression | Skn-1_motif | GTCAT | −221(−) −859(+) | −1366(+) −738(−) |
| meristem expression | CAT-box | GCCACT | −1648(+) | −145(−) |
| low-temperature responsiveness | LTR | CCGAAA | −1557(+) | −264(−) |
| defence and stress responsiveness | TC-rich repeats | ATTTTCTCCA | −1966(+) −1456(+) | −331(−) |

**Table 3.** Specific *cis*-acting elements in the promoter regions of *BaPDS1* and *BaPDS2*.

| promoter | function | *cis*-element | sequence (5′ – 3′) | location |
|---|---|---|---|---|
| *BaPDS1* | light responsiveness | GATA-motif | AAGGATAAGG | −1856(+) |
| | | TCCC-motif | TCTCCCT | −1341(+) |
| | | AE-box | AGAAACAA | −159(−) |
| | MeJA- responsiveness | CGTCA-motif | CGTCA | −35(−) |
| | | TGACG-motif | TGACG | −1970(+) |
| | ABA responsiveness | ABRE | TACGTG | −1615(+) −401(−) |
| | MYB binding site involved in drought-inducibility | MBS | TAACTG | −1754(+) −304(−) −1396(−) |
| | wound-responsiveness | WUN-motif | TCATTACGAA | −1153(−) |
| *BaPDS2* | light responsiveness | GT1-motif | AATCCACA | −349(−) |
| | | GA-motif | ATAGATAA | −951(−) −1757(+) |
| | | GAG-motif | AGAGAGT | −906(−) |
| | | AT1-motif | AATTATTTTTATT | −1461(+) |
| | | ATCT-motif | AATCTAATCT | −478(−) |
| | gibberellin responsiveness | GARE-motif | AAACAGA | −1777(+) |
| | MYB binding site involved in light responsiveness | MRE | AACCTAA | −836(−) |

results show that BaPDS1 and BaPDS2 were specifically localized in the chloroplast, which was consistent with the software prediction.

## 3.5. Phenotypic analysis of *BaPDSs* transgenic plants

Conserved target sites of BaPDS1 and BaPDS2 were selected using CRISPR DESIGN and ZiFiT Targeter v. 4.2 software. The target sequence was GATGGAGATTGGTATGAAAC**CGG**; with the PAM site shown in bold letters. The constructed recombinant plasmid and the empty vector were, respectively, transferred into the cotyledons of the Chinese kale using an *Agrobacterium*-mediated method, and transgenic plants were obtained after culture (figure 3). Monoclonal sequencing analysis showed that the transgenic plants were all homozygous for the mutation, and the mutation sites were all at the target site. M1 is a *BaPDS1* single mutant and a 3 bp replacement occurred at the target site. M2 is a *BaPDS2* single mutant with a 5 bp insertion and a 1 bp deletion at the target site. M3 is a mutant with *BaPDS1* and *BaPDS2* double mutations, with edits occurring in both genes. Nucleotide changes also cause changes in amino acid sequences. Point mutations of M1 and M3 caused premature termination of translation at the target site, while point mutations and deletions of M2 caused changes in amino acids and premature termination at the 94 base after the target sequence. In conclusion, premature termination of translation resulted in the loss of gene function in all three mutants (figure 4). At the same time, it was found that the leaves of this double mutant were completely albino compared to wild-type and empty vector plants, while the leaves of *BaPDS1* and *BaPDS2* single mutants were both only partially albino.

The colour results (table 4) showed that the red/green (*a*\*) values of the leaves of *BaPDS* single and double mutants were all significantly higher than those of the wild-type and empty vector plants, and no significant difference of *a*\* value was observed between the three mutants. The yellow/blue (*b*\*) values were not significantly different between the plants tested, except for the *BaPDS1* single mutant. In addition, the brightness (*L*\*) values of the mutants were not significantly different from the control. In summary, plants with double mutations of *BaPDS1* and *BaPDS2* tended to be whitened, while plants with single mutations of *BaPDS1* or *BaPDS2* were more prone to yellowing. However, to confirm the differences, multiple independent single- and double-mutant lines need to be assessed in the future.

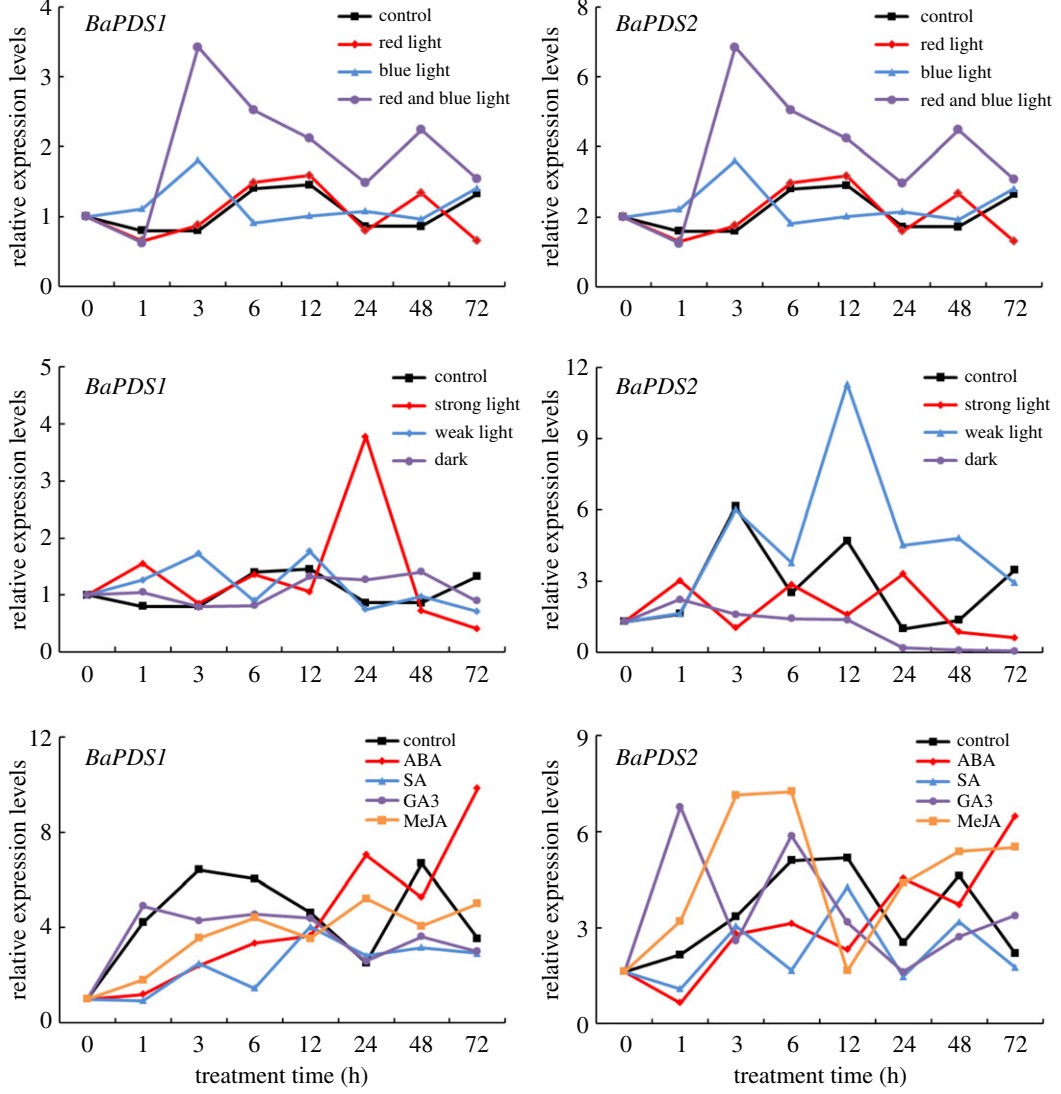

**Figure 1.** Gene expression of *BaPDS1* and *BaPDS2* under light and phytohormone treatments.

## 4. Discussion

Light is a stimulus that activates a broad range of plant genes that are related to both photosynthesis and photomorphogenesis, as well as biosynthetic genes involved in carotenoid biosynthesis [20]. Many studies reported that light quality can remarkably influence carotenoid accumulation and relative *PDS* gene expressions with the underlying factors varying between genes and across species. For example, increased transcript levels for carotenoid biosynthesis genes in green alga were detected under both blue and red light conditions [21]. The accumulation of β-cryptoxanthin in citrus fruits was induced by red light, while it was not affected by blue light [22]. In *Chlamydomonas reinhardtii*, only blue light was effective, whereas illumination with red light did not lead to elevated transcript levels of phytoene synthase (PSY) and PDS [23]. By contrast, the expression levels of carotenoid biosynthetic genes and the accumulation of total carotenoids were clearly lower in tartary buckwheat sprouts irradiated with blue and red light than in those irradiated with white light [24]. In this experiment, red and blue and blue light could induce *BaPDS1* in the seedlings of Chinese kale, while blue light could suppress *BaPDS2*, indicating that both genes have different response mechanisms under distinct light qualities, which is similar to the results of previous studies. Carotenoid accumulation and the genes controlling carotenoid biosynthesis are also regulated by light intensity. Astaxanthin accumulation and the gene expression levels of *PSY* and *PDS* in green alga were inhibited under low-light conditions [21], and a rapid decrease in the *PDS* transcripts in *C. reinhardtii* was observed both under high light and in the dark [23]. In this study, the expression level of *BaPDS1* peaked after being

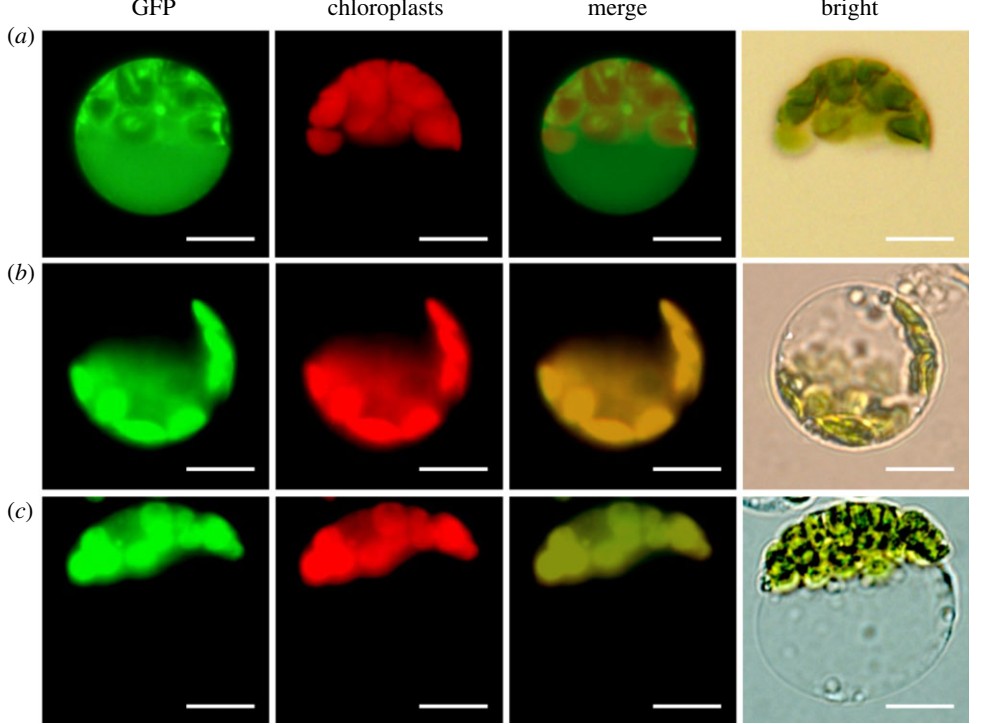

**Figure 2.** Subcellular location of BaPDS1 and BaPDS2. (*a*) Transient expression of GFP protein in Chinese kale protoplasts; (*b*) transient expression of BaPDS1-GFP fusion protein in Chinese kale protoplasts; (*c*) transient expression of BaPDS2-GFP fusion protein in Chinese kale protoplasts; bar, 10 μm.

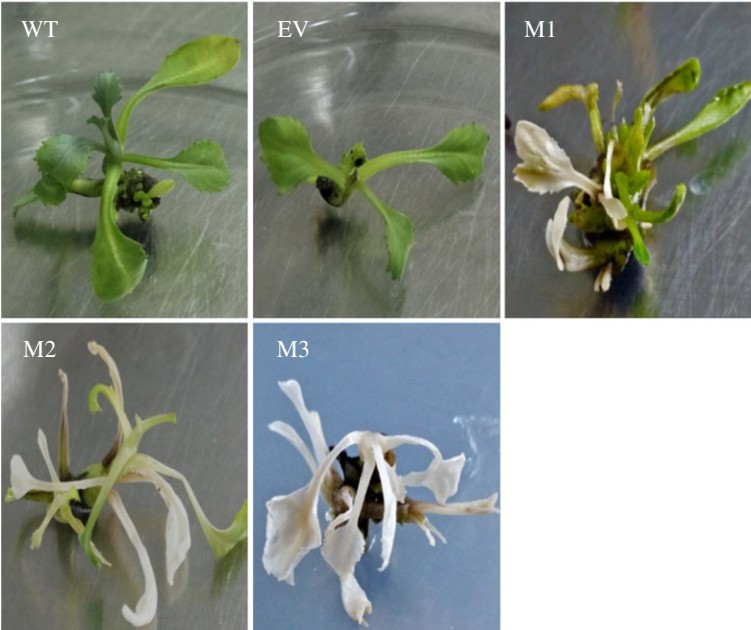

**Figure 3.** Phenotypes of the Chinese kale *BaPDS* mutants and control plants. WT, wild-type plant; EV, transgenic plants with empty vector; M1, *BaPDS1* and *BaPDS2* double mutant; M2, *BaPDS1* single mutant; M3, *BaPDS2* single mutant.

exposed to strong light for 24 h, which may be in response to chlorophyll damage. Carotenoids play an important role in the photosynthesis. They not only act as photosynthetic antenna-assisted pigments for absorbing and transforming light energy [25], but also protect chloroplasts from damage by absorbing residual light energy [26]. As a key rate-limiting enzyme in the carotenoid biosynthesis pathway, enhanced expression of *BaPDS1* might reduce chlorophyll damage. However, under low-light

BaPDS1 target site

BaPDS2 target site

**Figure 4.** Mutation detection in Chinese kale *BaPDS* mutants mediated by the CRISPR/Cas9 system. WT, wild-type plant; M1, *BaPDS1* single mutant; M2, *BaPDS2* single mutant; M3, *BaPDS1* and *BaPDS2* double mutant. The target sequence is indicated in blue, the PAM sequence (NGG) is underlined in red, mutated bases are indicated in red font and the short line represents the deletion base. r #, # number of base replacements; d #, # number of base deletions.

**Table 4.** The colour of leaves of the *BaPDS* mutants and control plants. Data are expressed as mean standard deviation. Same letter in the same column means no significant differences among values ($p < 0.05$) according to the LSD's test.

| plant no. | mutation type | $L*$ | $a*$ | $b*$ |
|---|---|---|---|---|
| WT | wild-type | $33.52 \pm 0.36^a$ | $-1.43 \pm 0.04^b$ | $2.10 \pm 0.27^b$ |
| EV | empty vector | $33.24 \pm 0.66^a$ | $-1.35 \pm 0.08^b$ | $2.10 \pm 0.01^b$ |
| M1 | *BaPDS1* mutant | $34.23 \pm 0.36^a$ | $-0.07 \pm 0.26^a$ | $2.65 \pm 0.03^a$ |
| M2 | *BaPDS2* mutant | $33.62 \pm 0.95^a$ | $0.11 \pm 0.10^a$ | $2.31 \pm 0.16^{a,b}$ |
| M3 | double mutant | $34.27 \pm 0.28^a$ | $-0.08 \pm 0.02^a$ | $1.88 \pm 0.13^b$ |

conditions, the chlorophyll content in plants decreased, and the expression of *BaPDS2* was upregulated, which could help chlorophyll to better absorb light energy. In short, these results indicate that both *BaPDS1* and *BaPDS2* in Chinese kale have their own response to different light qualities and light intensities, reflecting their distinct response to light exposure.

Classic phytohormones are involved in the regulation of carotenoid accumulation in plants [27]. SA, which is a plant hormone synthesized by plants, has been found to enhance disease resistance by regulating photosynthesis [28,29]. SA response elements were found in the 5′ upstream flanking sequence of *PDS* in *Scenedesmus obliquus*, and the *SoPDS* gene was highly expressed after 3.5 mg l$^{-1}$ SA treatment [30]. A twofold increase in chlorophyll content and a 3.5-fold increase in carotenoid content were observed following treatment with 10 µM or 0.1 µM SA, respectively, in sunflower cotyledons compared to the control [28]. By contrast, the lycopene content in tomato fruit declined after treatment with 5 µM SA [29]. In this study, SA treatment inhibited the expression of both *BaPDS* genes in Chinese kale. This difference compared to tomato may be due to the SA concentrations that were applied or cross-species differences in response to this hormone. ABA is a compound with a semi-quinone structure that is indirectly synthesized by carotenoid degradation in plants, and has a regulatory effect on seed germination, plant growth and senescence. In 'MicroTina' tomato plants treated with ABA, a significant increase in β-carotene, lutein, zeaxanthin and neoxanthin was observed in the leaf tissue, and in the lycopene level in fruit tissue [31]. In the present study, the expression levels of *BaPDS1* and *BaPDS2* in Chinese kale seedlings increased gradually after ABA treatment, which is consistent with previous studies. GA$_3$ is the major hormone regulating plant growth and cell division. Its synthetic precursor in plants is geranylgeranyl pyrophosphate (GGPP), which is the same as the synthetic precursor of carotenoids. Exogenous application of a high concentration of GA3 reduced the accumulation of lycopene and β-carotene in the flesh of the Cara Cara oranges, while application of a low concentration (50 mg l$^{-1}$) facilitated the accumulation of lycopene [32]. Similar results were also found in our study, and GA$_3$ at 5 µM (which is a relatively low concentration) quickly increased *BaPDS2* expression. MeJA is often used as a signalling molecule to transmit signals to plants, and induce a defence mechanism against various stresses. After fumigation of Chinese kale seedlings with MeJA, the expression level of *BaPDS2* was upregulated, indicating that this gene is likely involved in the response to MeJA against external stresses.

In summary, both *BaPDS* genes in Chinese kale showed similar responses to SA and ABA, while only *BaPDS2* could respond to GA$_3$ and MeJA, perhaps implying that *BaPDS2* plays a more important role in the phytohormone response.

Transcriptional regulation plays an important role in the regulation of gene expression, and is mainly controlled by gene promoters and their contributing *cis*-acting elements that are located upstream of the transcriptional start site [33,34]. A number of algorithms and bioinformatics tools have been developed to identify potential *cis*-acting elements [35]. The promoter regions of auxin response factor (ARF) genes in barley were analysed, and the differences of *cis*-acting elements between promoters of this gene family were investigated [36]. A diversity of the stress-related *cis*-acting elements were found in the 46 pyrabactin resistance 1-like (*PYL*) promoters in *Brassica napus*, and the gene expression patterns of a total of 14 *BnPYL* genes under relative abiotic stresses were investigated [12]. To understand the respective roles of *BaPDS1* and *BaPDS2*, in this study, the differences in *cis*-elements between both *BaPDS* promoters were analysed, with the conserved and differing *cis*-elements being mainly associated with light and hormone responsiveness. Therefore, in Chinese kale seedlings, light quality, light intensity and phytohormones were used as the basis for comparing gene expression. This result showed that the specific gene expression of *BaPDSs* under light, SA and GA$_3$ treatments were correlated with the predicted *cis*-acting elements, demonstrating the relevance of selecting treatments based on predicted *cis*-acting elements for these factors. However, this correlation did not hold true for *cis*-elements responsive to ABA and MeJA. This may be caused by the missing potential *cis*-acting elements located in the distal promoter region, or inter-species differences between Chinese kale and the reference species in the database [33], while additional research correlating *cis*-acting elements with treatments as well as advances in bioinformatics may resolve such apparent discrepancies.

The function of a protein is typically closely related to both its structure and location. After the protein matures, it must be correctly localized to exert its function. The subcellular localization information of the protein strongly suggests its biological function. Therefore, subcellular localization studies are a key step towards revealing protein function. In our previous study, an efficient isolation and transformation system of Chinese kale mesophyll protoplasts was established, and BaPDS1 in Chinese kale was found to target the chloroplast [18]. This experiment analysed the localization of BaPDSs by both bioinformatics analysis and subcellular localization experiments, and showed that BaPDS1 and BaPDS2 were both localized in the chloroplast, which is consistent with the results that indicate that PDS in *Haematococcus pluvialis* was localized in chloroplasts using immunogold localization [37]. This result indicates that BaPDS1 and BaPDS2 are both involved in the photosynthesis of Chinese kale.

The CRISPR/Cas9 system is a next-generation genome editing tool that has been increasingly applied to a variety of plants [11]. Previously, CRISPR/Cas9 technology was used for site-directed mutagenesis on several multicopy genes, such as the granule-bound starch synthase (GBSS) genes of tetraploid potato [38], the multi-chamber developmental gene CLAVATA3 of *B. napus* [39] and the delta-12 desaturase gene (FAD2) of *Camelina sativa* [40]. In this study, a conserved target site was used to edit both *BaPDS* genes using the CRISPR/Cas9 system, and single mutants of *BaPDS1* and *BaPDS2*, as well as the double mutant, were simultaneously obtained. The phenotype of the mutants showed that the double mutant was more completely whitened than each single mutant, while the single mutant plants were both more inclined to yellowing (figure 3 and table 4). This indicates that the roles of *BaPDS1* and *BaPDS2* in carotenoid metabolism of Chinese kale are distinct, partially complementary, equally important, with the double mutant being white, likely compromising its photosynthetic performance.

## 5. Conclusion

The promoters of two *PDS* gene family members of the Chinese kale (*BaPDS1* and *BaPDS2*) were cloned in this study. The results indicate that the promoters of both genes contained many common and different *cis*-acting elements. Gene expression results showed that *BaPDS1* could be induced by red and blue, blue and strong light, while *BaPDS2* could be induced by weak light, GA$_3$ and MeJA, and suppressed by blue light and dark. Both *BaPDS1* and *BaPDS2* responded to ABA and SA. Subcellular localization indicated that both BaPDS1 and BaPDS2 were localized in the chloroplast. Moreover, *BaPDS* genes were edited using the CRISPR/Cas9 technology and neither single mutant was albino, whereas the leaves of the double mutant were completely white. These findings demonstrate that *BaPDS1* and *BaPDS2* could be partially redundant during the growth and development of Chinese kale.

Data accessibility. All data used in this manuscript are present in the manuscript.

Authors' contributions. H.-R.T., F.Z. and B.S. conceived and designed the experiments; M.J., S.L., H.Z., Q.C. and Y.W. performed the experiments; B.S., Y.-X.L., Z.-J.L. and X.R.W. analysed the data; B.S. and M.J. wrote the paper. All authors gave final approval for publication.

Competing interests. We declare we have no competing interests.

Funding. This work was supported by National Natural Science Foundation of China (31500247), Sichuan Science and Technology Program (2018NZ0081), National Student Innovation Training Program (201710626030) and Undergraduate Research Interest Cultivation Project of Sichuan Agricultural University (2019303, 2019304).

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
