## [Reviewer comments · Royal Society Open Science]

Review History

RSOS-190260.R0 (Original submission)

Review form: Reviewer 1

Is the manuscript scientifically sound in its present form?

Yes

Are the interpretations and conclusions justified by the results?

Yes

Is the language acceptable?

Yes

Is it clear how to access all supporting data?

Yes

Do you have any ethical concerns with this paper?

No

Have you any concerns about statistical analyses in this paper?

No

Recommendation?

Accept with minor revision (please list in comments)

Comments to the Author(s)

Comments to the Author(s)

This is an acceptable paper for publication in Royal Society Open Science. The manuscript entitled "Functional Differences of BaPDS1 and BaPDS2 Genes in Chinese Kale" is an interesting and nice work. The manuscript reported the differences of promoter cis-elements, expression patterns, and subcellular localization between the two genes. The results also well explained the functional differences via CRISPR/Cas9 gene editing. This work will provide a reference for the functional research between various members of a gene family.

However, some comments should be made to improve the manuscript. Below are some questions and/or suggestions.

1) M&M

Does the light intensity mean the light source itself? If yes, how much is it at the surface of plant leaves? How much did you spray for each phytohormes?

2) Fig 1

According to your analysis, there are MeJA and ABA response cis-element in the promoter of PDS1. However, PDS2 shows stronger response to the MeJA treatment. Could you discuss the possible cause?

3) Fig 2A

The 3rd image should be the merge of red and green. However, the red channel can hardly be seen. Could you check the parameter?

4) Fig 4

The authors showed the DNA sequence of the edited plants. However, further analysis, such as the effect of nucleotide sequence change on the translation or protein sequence should be done.

Review form: Reviewer 2**Is the manuscript scientifically sound in its present form?**

Yes

Are the interpretations and conclusions justified by the results?

Yes

Is the language acceptable?

No

Is it clear how to access all supporting data?

Not Applicable

Do you have any ethical concerns with this paper?

No

Have you any concerns about statistical analyses in this paper?

I do not feel qualified to assess the statistics

Recommendation?

Major revision is needed (please make suggestions in comments)

Comments to the Author(s)

This paper shows a single, presumably loss of function gene edited line for each of two phytoene desaturase homologs as well as a double mutant in Chinese kale. The authors characterize the response of these genes to various light qualities, intensities as well as plant hormones and relate these to cis element in each of the two promoter sequences. Overall the quality of the manuscript could have been better as evidenced by the large number of recommended edits. In addition, the paper would have been stronger if the phenotypes of multiple independent single and double mutant lines were assessed. The partial redundancy of these genes, characterization of cis elements in promoters for these genes combined with gene expression under various light and hormonal treatments makes for an interesting story.

Decision letter (RSOS-190260.R0)

14-May-2019

Dear Dr Tang,

The editors assigned to your paper ("Functional Differences of BaPDS1 and BaPDS2 Genes in Chinese Kale") have now received comments from reviewers. We would like you to revise your paper in accordance with the referee and Associate Editor suggestions which can be found below (not including confidential reports to the Editor). Please note this decision does not guarantee eventual acceptance. Please ensure that, as well as addressing the reviewer and Editor concerns, you seek professional language editing advice (<https://royalsociety.org/journals/authors/language-polishing/>).

Please submit a copy of your revised paper before 06-Jun-2019. Please note that the revision deadline will expire at 00.00am on this date. If we do not hear from you within this time then it will be assumed that the paper has been withdrawn. In exceptional circumstances, extensions may be possible if agreed with the Editorial Office in advance. We do not allow multiple rounds of revision so we urge you to make every effort to fully address all of the comments at this stage. If deemed necessary by the Editors, your manuscript will be sent back to one or more of the original reviewers for assessment. If the original reviewers are not available, we may invite new reviewers.

- Data accessibility

If you wish to submit your supporting data or code to Dryad (<http://datadryad.org/>), or modify your current submission to dryad, please use the following link:
<http://datadryad.org/submit?journalID=RSOS&manu=RSOS-190260>

- Competing interests

- Authors' contributions

- Acknowledgements

- Funding statement

Kind regards,

Andrew Dunn

on behalf of Dr Stephen Long (Associate Editor) and Catrin Pritchard (Subject Editor)

Associate Editor's comments (Dr Stephen Long):

This is a sound piece of science, that adds interesting information on independent roles, both essential, for two gene family members. Reviewer #1 raises key issues that must be addressed through corrections/additions to the ms. The ms is poorly written at many points and that has to be addressed if the ms is to become finally acceptable for publication. Reviewer #2 has noted many points in the English that need revision/correction. It is essential that all are attended to, and the final version is reviewed with a native English speaker before final submission. The final submission must be accompanied by a letter, explaining how each point in the reviewers' reports are addressed.

Comments to Author:

Reviewers' Comments to Author:

Reviewer: 1

Comments to the Author(s)

Comments to the Author(s)

This is an acceptable paper for publication in Royal Society Open Science. The manuscript entitled "Functional Differences of BaPDS1 and BaPDS2 Genes in Chinese Kale" is an interesting and nice work. The manuscript reported the differences of promoter cis-elements, expression patterns, and subcellular localization between the two genes. The results also well explained the functional differences via CRISPR/Cas9 gene editing. This work will provide a reference for the functional research between various members of a gene family.

However, some comments should be made to improve the manuscript. Below are some questions and/or suggestions.

1) M&M

Does the light intensity mean the light source itself? If yes, how much is it at the surface of plant leaves? How much did you spray for each phytohormes?

2) Fig 1

According to your analysis, there are MeJA and ABA response cis-element in the promoter of PDS1. However, PDS2 shows stronger response to the MeJA treatment. Could you discuss the possible cause?

3) Fig 2A

The 3rd image should be the merge of red and green. However, the red channel can hardly be seen. Could you check the parameter?

4) Fig 4

The authors showed the DNA sequence of the edited plants. However, further analysis, such as the effect of nucleotide sequence change on the translation or protein sequence should be done.

Reviewer: 2

Comments to the Author(s)

This paper shows a single, presumably loss of function gene edited line for each of two phytoene desaturase homologs as well as a double mutant in Chinese kale. The authors characterize the response of these genes to various light qualities, intensities as well as plant hormones and relate these to cis element in each of the two promoter sequences. Overall the quality of the manuscript could have been better as evidenced by the large number of recommended edits. In addition, the paper would have been stronger if the phenotypes of multiple independent single and double mutant lines were assessed. The partial redundancy of these genes, characterization of cis elements in promoters for these genes combined with gene expression under various light and hormonal treatments makes for an interesting story.

Author's Response to Decision Letter for (RSOS-190260.R0)

See Appendix A.

Decision letter (RSOS-190260.R1)

14-Jun-2019

Dear Dr Tang,

I am pleased to inform you that your manuscript entitled "Functional Differences of BaPDS1 and BaPDS2 Genes in Chinese Kale" is now accepted for publication in Royal Society Open Science.

on behalf of Dr Stephen Long (Associate Editor) and Catrin Pritchard (Subject Editor)
openscience@royalsociety.org

Associate Editor Comments to Author (Dr Stephen Long):

The authors have satisfactorily attended to all the issues that raised by myself and the two referees. This paper is now ready for publication.

Appendix A

Manuscript ID: RSOS-190260

Title: Functional Differences of *BaPDS1* and *BaPDS2* Genes in Chinese Kale

Dear Professor Catrin Pritchard

We are pleased to submit a revised version of the manuscript RSOS-190260. We are truly grateful to editor and reviewers' critical comments and thoughtful suggestions. We have worked through the details in the comments of the reviewers, and revised the paper accordingly. We hope the new manuscript will meet the standard of *Royal Society Open Science*. Below you will find our point-by-point responses to the reviewers' comments/questions.

Thank you very much for your consideration.

Yours sincerely,

Hao-Ru Tang

Editor' Comments to Author:

1) Reviewer #1 raises key issues that must be addressed through corrections/additions to the ms. The ms is poorly written at many points and that has to be addressed if the ms is to become finally acceptable for publication.

Thanks for your suggestion. We have made corresponding revises according to the opinions of reviewer 1.

1) Reviewer #2 has noted many points in the English that need revision/correction. It is essential that all are attended to, and the final version is reviewed with a native English speaker before final submission.

Thanks for your suggestion. We have revised point by point according to the opinions in the document of reviewer 2. We also have carefully checked and revised the manuscript to improve the English expression by ourselves, and then the manuscript was also critically read by Dr. Yunting Zhang (University of California, Davis).

Referees' Comments to Author:

Referee: 1

1) M&M

Does the light intensity mean the light source itself? If yes, how much is it at the surface of plant leaves? How much did you spray for each phytohormes?

It is a very good suggestion. The light intensity means the light source itself. The light source is directly above the plant, and the linear distance from light to the plants is about 15 cm. We have stopped spraying until the leaf surface was dripping, and the amount was about 2 mL of each phytohormones per plant. We have revised them in the manuscript.

2) Fig 1

According to your analysis, there are MeJA and ABA response cis-element in the promoter of PDS1. However, PDS2 shows stronger response to the MeJA treatment. Could you discuss the possible cause?

Thanks for your suggestion. The results of gene expressions under ABA and MeJA treatments differed from the predicted results. This may be caused by the missing potential cis-acting elements located in the distal promoter region, or the inter-species differences between Chinese kale and the reference species in the database, while additional research results of cis-acting elements and advances in bioinformatics will accelerate the resolution of this problem in the future.

3) Fig 2A

The 3rd image should be the merge of red and green. However, the red channel can hardly be seen. Could you check the parameter?

It is a very good suggestion. We have adjusted the parameter of Figure 2A, and revised it in the manuscript.

4) Fig 4

The authors showed the DNA sequence of the edited plants. However, further analysis, such as the effect of nucleotide sequence change on the translation or protein sequence should be done.

It is a very good suggestion. We have analyzed the effect of nucleotide changes on translation and protein, and added the results in Figure 4 and the manuscript.

Referee: 2

1) Overall the quality of the manuscript could have been better as evidenced by the large number of recommended edits.

Thanks for your suggestion. We have revised point by point according to the opinions in your document. We also have carefully checked and revised the

manuscript to improve the English expression by ourselves, and then the manuscript was also critically read by Dr. Yunting Zhang (University of California, Davis).

2) p2 line 30 – recommend adding cassava - Odipio J, Alicai T, Ingelbrecht I, Nusinow DA, Bart R, Taylor NJ. 2017 Efficient CRISPR/Cas9 genome editing of phytoene desaturase in cassava. *Frontiers in Plant Science* 18, 1780.

It is a very good suggestion. This is a very good article enlightening to our experiment, and we have quoted this article in our manuscript.

3) p3 line 37 – What is meant by an artificial intelligence incubator?

Thanks for your suggestion. Actually, the artificial intelligence incubator is an artificial climate chamber that can control the conditions of temperature, humidity and light. We have revised it in the manuscript.

4) In addition, the paper would have been stronger if the phenotypes of multiple independent single and double mutant lines were assessed.

It is a very good suggestion. As suggested, we will obtain more single and double mutant plant lines in later experiments to confirm our results.